# Estimation of Ecological and Human Health Risks Posed by Heavy Metals in Street Dust of Madrid City (Spain)

**DOI:** 10.3390/ijerph19095263

**Published:** 2022-04-26

**Authors:** María José Delgado-Iniesta, Pura Marín-Sanleandro, Elvira Díaz-Pereira, Francisco Bautista, Miriam Romero-Muñoz, Antonio Sánchez-Navarro

**Affiliations:** 1Department of Agricultural Chemistry, Geology and Pedology, Faculty of Chemistry, Campus de Espinardo, University of Murcia, 30100 Murcia, Spain; delini@um.es (M.J.D.-I.); miriam.romero1@um.es (M.R.-M.); antsanav@um.es (A.S.-N.); 2Soil and Water Conservation Research Group, Spanish National Research Council (CEBAS-CSIC), Campus de Espinardo, 30100 Murcia, Spain; ediazpereira@cebas.csic.es; 3University Laboratory of Environmental Geophysics (LUGA), Environmental Geography Research Center, National Autonomous University of Mexico, Mexico No. 8701, Morelia 58190, Michoacan, Mexico; leptosol@ciga.umam.mx

**Keywords:** urban dust, urban pollution, hazard index, carcinogenic risk, ecological risk index

## Abstract

In this work, sampling was carried out in the urban area of Madrid to analyze the content of total heavy metals (Zn, Pb, Cu, Cr, Ni, and Cd) in the street dust. Contamination was evaluated using various indices, such as the Contamination Factor (CF), Enrichment Factor (EF), Geo-accumulation Index (Igeo), Potential Ecological Risk Index (RI), Pollution Load Index (PLI), the Human Health Index Hazard Index (HI), and Cancer Risk (CR). Pollution indices were related to traffic density and color. Traffic density was the factor that most influenced the values of the pollution indexes, but no significant differences were found with the color of street dust. The concentration of heavy metals in the urban dust of Madrid had the following sequence: Zn (895) > Cu (411) > Pb (290) > Cr (100) > Ni (42) > Cd (1.25 mg kg^−1^). The pollution levels were high or very high in Pb, Zn, and Cd regarding the environmental pollution indexes. Ingestion was the main route of exposure to heavy metals contained in street dust. The CR for adults and children is less than 1 × 10^−6^, which means that there is no risk for the population. However, the HI was 10 times higher in children than in adults.

## 1. Introduction

Street dust is a heterogeneous mixture of particles from natural and anthropogenic sources. Among the natural sources, erosion, weathering, resuspension, and deposition of soil particles are considered the primary street dust sources. In contrast, emissions from several urbanization activities such as traffic, industrial and household emissions, and wear and tear on the built environment, such as houses, roads, and buildings, are considered primary anthropogenic sources. Heavy metals have great ecological significance due to their toxicity and tendency to accumulate in sediment, soils, and biota. These elements are not biodegradable and undergo a global ecological cycle [1].

Heavy metal contamination in street dust varies according to the type of activity [2], and there are studies of street dust under different uses, such as industrial, mining, residential, commercial, urban, and agricultural [3,4,5], and with varying intensities of traffic [6]. There is particular interest in studying gardens and parks where children play [7,8]. Street dust particles can reach rivers through sewage, increasing their contamination, as reported when analyzing urban and road runoff [9].

Pollution from particle matter is considered a severe threat, especially for numerous metals such as Fe, Cu, Zn, and Pb, generated through emissions. Cars are the key contributor to the resuspension of street dust, heavy metals, and other carcinogenic substances [10]. The association of detrimental health effects with pollutants present in urban areas is a serious issue that residents face.

Street dust is a source of metal pollutants that can damage human health through ingestion, dermal contact, or inhalation, being more harmful the smaller the particle size [8,11].

Many people who live or carry out their daily activities in urban centers suffer from health problems due to continuous exposure to heavy metals and other pollutants [12,13,14]. Heavy metals may accumulate in the fatty tissues and affect the central nervous system, be deposited in the circulatory system, and disrupt internal organs’ normal functioning. This can cause DNA damage and have mutagenic, teratogenic, and carcinogenic effects. Mainly, As, Cd, and Cr have been related to various types of cancer [15], and high blood Pb levels could affect the nervous system and brain tissue in human bodies.

Various indices are used to assess the environmental and health risk caused by exposure to heavy metals in street dust [16,17,18,19,20]. They are very suitable tools that allow for knowing the situation and acting in case of risk.

In recent decades, the study of the concentration of heavy metals in street dust in cities has aroused growing interest and thus, has been studied in highly populated capitals such as Mexico City [21,22], Dakha [23], Islamabad [24], and Istanbul [25]. Madrid was studied by [7,26] and compared with Oslo to discover the differences in the origin and nature of street dust in these two capitals with very different lifestyles.

The Barcelona Institute for Global Health recently reported that the mortality burden attributable to air pollution in cities was due to NO_2_ and PM 2.5; Madrid ranked 1 and 551, respectively [27]. The PM2.5 particles are street dust components. The population of Madrid has grown in the last two decades with a consequent increase in air pollution due to more people, cars, factories, and more emissions into the atmosphere. The pollution has exceeded the levels considered permissible, forcing the political authorities to take urgent measures such as restricting the circulation of cars with even or odd number plates and reducing speed on some critical roads. Madrid Central has been a low emissions zone (LEZ) since November 2018, when the Air Quality and Climate Change plan prohibited traffic from passing through the city center. This has improved the current environmental situation considerably [28]. On 22 September 2021, the new sustainable mobility ordinance came into force, essentially affecting the Special Protection LEZ of the Central District, which maintains the same restrictions, meaning only cars with an ECO and ZERO label can circulate.

The hypothesis is that street dust in the city of Madrid contains, among other pollutants, heavy metals, the concentration of which could depend on different factors, such as the density of traffic, and that these heavy metals could be one of the causes of the increase in pathologies, some very serious.

The aims of this study are: (a) to evaluate the concentration level of six common heavy metals (Cu, Pb, Zn, Cd, Ni, Cr) associated with traffic emissions and discover their evolution in the last two decades; (b) assess the level of pollution by calculating different indices; (c) test whether the concentration of heavy metals has any relationship with the color of street dust and with the density of traffic; (d) perform a health risk assessment for both children and adults according to hazard index (HI) and carcinogenic risk (CR) methods.

The novel contributions of this paper are as follows:

A systematic sampling, which allows us to know precisely the spatial distribution of the variables studied.

The inclusion of the urban dust color factor as a grouping variable.

The use of asphalt as a geochemical background, since most of the papers consulted used soils from the surrounding area as a background.

## 2. Materials and Methods

### 2.1. Study Area and Sampling

The Madrid Metropolitan Area, which extends to over 600 km^2^, is the largest urban agglomeration in Spain, with approximately 6 million inhabitants. The climate in the Madrid region is continental with moderately cold winters and hot summers (annual average temperature 14.6 °C) and relatively dry with an average precipitation of 436 mm year^−1^. These conditions and traditional Spanish social habits encourage outdoor activities for people, particularly children in playgrounds after school. The predominant soils in the area around the city are mainly Entisols, Inceptisols, Alfisols, and Mollisols [29].

The study area was the urban center of Madrid, currently known as the low emission zone (LEZ). It has an area of 7.5 km^2^, and it is the prototype of the European capital. The street dust sampling was carried out in November 2016 before the perimeter closure of traffic in this area of Madrid.

The sampling design was systematic, with 500 m between site samples, taking 35 street dust samples (Figure 1). The location of the samples, indicating their latitude and longitude, are presented in Appendix A.

The dust samples were taken by sweeping an area of 1 m^2^, following the procedures described in document USEPA AP-42 [30,31]. The dust samples were kept in plastic containers and subsequently sieved with a 1 mm light non-metallic sieve.

We consider it convenient to analyze the concentration of heavy metals in the asphalt with the idea of taking them into account in the evaluation of street dust contamination. Fifteen asphalt samples were taken from the road using a hammer drill, collecting the dust in plastic containers.

### 2.2. Street Dust Analysis Study Area and Sampling

The color of the samples was estimated with the Munsell colors chart [32], and then they were clustered into two different categories: lights (value between 5 and 6) and darks (value of 2, 3, and 4); two types had the same HUE 2.5Y. Daily traffic density was provided by the Área de Gobierno de Medio Ambiente y Movilidad. It was classified into three categories: low density, from 1000 to 10,000 vehicles/day; medium, from 10,000 to 40,000, and high, from 40,000 to 80,000 vehicles/day.

The samples were air-dried and passed through a 50 µm sieve prior to acid digestion with aqua regia (HNO_3_/HCl, 1:3) in a microwave oven at 220 °C for 1 h [33]. Subsequently, inductively coupled argon plasma mass spectrometry (ICP-MS model Agilent 7900) was used to determine the following elements: Cd, Cu, Cr, Ni, Pb, and Zn (total metal concentration).

Blank samples, a standard reference material certified for its element content (SRM San Joaquín Soil, reference NIST2709A, Sigma-Aldrich (St. Louis, MO, USA)), were used to provide the baseline trace element concentrations and duplicated samples were analyzed simultaneously to provide quality control. The standard deviation was calculated (2.5–3%) and can be considered satisfactory for environmental analysis. We obtained 93–102% recoveries for Zn, 95–101% for Pb, 91–98% for Cu, 94–99% for Cr, 98–102% for Ni, and 95–101% for Cd.

X-ray diffraction (XRD) was used to detect the mineral composition of the samples. The samples were ground in an agate mortar and mounted on a petrographic slide prior to XRD analysis. The working conditions were radiation Kα Cu; intensity 24 mA; 40 kV; Ni filter; window slit of 1; meter gap of 0.1; scanning speed, 1° 2Θ min^−1^; and sensitivity, 5 × 10^3^ with Philips equipment with vertical goniometer.

### 2.3. Environmental Pollution Index and the Potential Risk

#### 2.3.1. Contamination Factor (CF)

The Contamination Factor (Equation (1)) describes the pollution level of street dust with a given heavy metal, and it was calculated as the ratio between the concentration of each heavy metal measured (*Cn*) and its background value (*Cbn*) [16,17].
(1)$$\mathrm{CF}=\frac{Cn}{Cbn}$$

Based on the results obtained for CF, the level of heavy metal contamination was established according to CF < 1, low; 1 ≤ CF ≤ 3, moderate; 3 ≤ CF < 6, considerable; and CF ≥ 6, very high.

#### 2.3.2. Pollution Load Index (PLI)

The Pollution Load Index (Equation (2)) is a tool used to assess the global level of sediment contamination, taking into account the concentrations of several heavy metals. PLI was calculated based on the Contamination Factor of each metal [18].
PLI = (*CF_Me*1*_ × CF_Me*2*_ × CF_Me*3*_ × … × CF_Men_*)^1*/n*^(2)
where PLI is the Pollution Load Index, *CF_Me_*_1,2,*…n*_ represents the Contamination Factor of each metal, and n is the number of metals. The values of PLI < 1 indicate the absence of heavy metal contamination, whereas PLI > 1 shows the presence of heavy metal pollution.

#### 2.3.3. Enrichment Factor (EF)

Enrichment Factor was initially proposed to describe metal pollution levels in the atmosphere [34,35]. Later on, this method was applied in many areas, especially for evaluating metals in soil and dust. EFs for metals in dust were calculated as follows (Equation (3)) [35,36]:(3)$$\mathrm{EF}=\frac{\left(\frac{Cx}{Cref}\right)sample}{\left(\frac{Cx}{Cref}\right)background}$$
where *Cx* represents the concentration of metal element *x*, and *Cref* is the concentration of the reference element—in our case, Ca, which is the major element in both dust and asphalt and no human activity influences. The background values used were those obtained after the analysis of the asphalt.

EF categories were defined as [36] deficient to minimal enrichment when EF < 2, moderate enrichment when EF = 2–5, substantial enrichment when EF = 5–20, very high enrichment when EF = 20–40, and extremely high enrichment when EF > 40.

#### 2.3.4. Geo-Accumulation Index (Igeo)

To obtain another contamination index to produce a more robust contamination degree analysis, the geo-accumulation Index (Igeo) was also calculated (Equation (4)). Igeo considers small variations in the background value using a 1.5 factor (factor K). The geo-accumulation index was proposed by [19] to assess the pollution levels of each heavy metal in surface sediments to account for their background value:(4)$$\mathrm{Igeo}=log2\frac{Cn}{K\times Cbn}$$
where *Cn* is the concentration of metal n and *Cbn* is the background concentration of the metal (*n*). The factor *K* is the background matrix correction factor due to lithospheric effects, which is usually defined as 1.5 [19].

Igeo can be interpreted as follows: Igeo < 0 (uncontaminated), 0–1 (uncontaminated to moderately contaminated), 1–2 (moderately contaminated), 2–3 (moderately to highlycontaminated), 3–4 (highly contaminated), 4–5 (highly to very highly contaminated), >5 (very highly contaminated).

#### 2.3.5. Potential Ecological Risk Index (RI)

The Potential Ecological Risk Index was developed by Häkanson [37] to evaluate the potential risk of heavy metal contamination in sediments (Equation (5)). According to Häkanson [37], the toxic response factors for heavy metals analyzed, such as Pb, Cu, Cr, Zn, and Ni, are 5, 5, 2, 1, and 5. The final value of RI was obtained by calculating the following formulas:RI = *ΣErMe*(5)
ErMe = *TrMe × CFMe*
where RI is the sum of potential risk of individual heavy metal, *ErMe* is the potential ecological risk of each metal *Me*, *TrMe* refers to the toxic response factor for each metal *Me*, *CFMe* is the contamination factor for each heavy metal.

The levels of ecological risk according to the RI index values obtained are RI < 150, low ecological risk; 150 ≤ RI < 300, moderate ecological risk; 300 ≤ RI <600, considerable ecological risk, and RI ≥ 600, very high ecological risk for the sediment.

### 2.4. Human Health Index and Cancer Risk

Human health risk assessment is a computational model to assess the likelihood of effects of hazardous chemicals on humans [20]. The human health risk assessment model was established by the United States Environmental Protection Agency and the Netherlands National Institute of Public Health [38,39,40]. In this model, citizens, including adults and children, can be exposed to heavy metals through the following three routes: ingestion (Ding), inhalation (Dinh), and dermal contact (Dder) [7,41].

In the first step for calculating the non-carcinogenic risk, the dose received through Ding, Dinh, and Dder was calculated using Equations (6)–(8) [38,39]. To assess the health risk from carcinogenic toxic metals, the lifetime median daily dose (LADDinh) for the inhalation exposure route of Cr, Cd, Ni, and Pb was applied using Equation (9) [39,42]:(6)$$\mathrm{Ding}=\frac{C\times ingR\times EF\times ED}{Bw\times AT}\times 10^{-6}$$
(7)$$\mathrm{Dinh}=\frac{C\times inhR\times EF\times ED}{PEF\times BW\times AT}$$
(8)$$\mathrm{Dder}=\frac{C\times SL\times SA\times ABS\times EF\times ED}{BW\times AT}\times 10^{-6}$$
(9)$$\mathrm{LADDinh}=\frac{C\times EF}{AT\times PEF}\times (\frac{inhRchild\times EDchild}{Bwchild}+\frac{inhRadult\times EDadult}{BWadult})$$
where the *IngR* intake rate is assumed to be 100 mg day^−1^ for adults, whereas it is 200 mg day^−1^ for children [42]; *InhR* as inhalation rate as inhalation rate is estimated to be 7.6 m^3^ day^−1^ for children and 20 m^3^ day^−1^ for adults [40], *C* is the content of toxic metals, mg kg^−1^; *EF* is the exposure factor, used in the present study of 350 days per year^−1^, *ED* is the exposure period, assumed in the present study of 6 years for children and 24 years for adults [42]; *SL* is the skin level, which in this study has been taken as 0.2 mg m^−2^ day^−1^ for children and 0.07 mg cm^−2^ day^−1^ for adults [42]; *SA* is the exposed skin area, estimated at 2800 cm^2^ for children and 5700 cm^2^ for adults [42]; *ABS* is the skin absorption factor, considered as 0.001 for all the heavy metals studied; *PEF* is the particulate emission factor = 1.36 × 10^9^ m^3^ kg^−1^ [42]; *AT* is the average time contact, defined *ED* × 365 days for non-cancers and 70 × 365 = 25,550 days for carcinogens [43], and *BW* is the mean body weight, defined as 15 kg for children and 70 for adults [42].

HI is the hazard index used to calculate the non-cancer risks of the metals studied for children and adults. The HI is the sum of the hazard quotients (HQ), which represents the magnitude of harmful impacts of the total exposure pathways (HI = *HQing + HQinh + HQder*) [44]. HQs for each exposure route were calculated by dividing the mean daily dose from each exposure route (Ding, Dinh, and Dder) by a corresponding reference dose (RfDing, RfDinh, and RfDder).

The RfD values (mg kg^−1^ day^−1^) illustrate the maximum permissible risk in the daily exposure of citizens throughout their lives [44]. The reference doses for the heavy metals studied are presented in Table 1. When HI < 1, it shows that there are no adverse health effects; when the HI is greater than 1, it indicates that probable non-cancer health impacts may occur [42].

Cancer Risk (CR) was defined as the probability of people developing cancer due to exposure to carcinogenic contaminants during their lifetime. Carcinogenic risks were calculated using Equation (10):Cancer Risk (CR) = LADDinh × SF(10)

For carcinogens, the dose is multiplied by the corresponding slope factor (SF) to produce an estimate of cancer risk. The SF of each carcinogenic element is shown in Table 1. Risk management decisions were most frequently made when the CR ranges were 10^−6^ to 10^−4^.

### 2.5. Statistical and Geostatistical Analysis

One-way ANOVA testing was performed, in which color of sample street dust and traffic density were the factors and heavy metal concentrations and pollution indices were the dependent variables (Duncan and Tukey) using IBM SPSS statics 28.0 software (SPSS Inc., Chicago, IL, USA). Statistical significance was set at *p* < 0.05.

The Pearson correlation matrix of the variable studied is included (Appendix A) to obtain the regression equation of the best correlated with each other.

Sampling sites and PLI values were viewed in Google Earth Pro to corroborate their spatial position.

PLI spatial distribution map was designed from a geostatistical analysis using the Geostatistics for the Environmental Sciences (GS+) software (Robertson 2021). The interpolation method was kriging indicator because it accepts data non-normality (Gaussian distribution) and converts the estimated values into indicator values that rank from 0 to 1 [21]. The output is the probability range of exceeding a cut-off value. We used a value of PLI = 20 as a cut-off value.

The ArcGIS 9.0 software was used for mapping. For a study of this scale level, we used the UTM projection, horizontal datum ellipsoid, and the World Geodetic System 84.

## 3. Results and Discussion

### 3.1. Composition of Street Dust

The minerals contained in the street dust from the city of Madrid are mainly calcite (C) and quartz (Q), with plagioclase and feldspar in smaller quantities (Figure 2).

It appears that metals such as Zn and Pb readily react with carbonates to form carbonate metal and mineral complexes on the surface of calcite or dolomite crystals [45].

The concentration of the six heavy metals analyzed differs depending on the element in question and ranges between mean values of 1.25 and 895 mg kg^−1^ for Cd and Zn, respectively (Table 2). On the other hand, there is also a great variability in the concentration of each of the elements analyzed, as found by other authors [4,46].

The results of this study were compared with those of similar studies carried out in other cities (Table 2), where there is also great variability in the mean values. If we compare Madrid, a large capital, with an average Spanish city such as Murcia, studied by Marín et al. [47], the PL value in Madrid is almost double that of Murcia, fundamentally attributable to the much higher Pb, Zn, and Cu contents, whereas the Ni and Cr contents are practically the same in both cities.

Regarding the evolution of pollution in the city of Madrid in the last 20 years, comparing the total load values of this work with those given by [26], the decrease in PL is attributable almost 80% to the reduction in Pb, clearly justified by the replacement of traditional fuels with lead-free fuels. However, Zn, Cu, and Cr are practically doubled in current values compared with 1997. Ni is the only element that remains at the same value.

Zn is the majority element in the street dust of Madrid, as in those analyzed in other works [22,23,24,25,47,48]. With a mean value of 895 mg kg^−1^ (Table 2), it is very similar to that found in other cities such as Paris [49]. The origin of Zn in street dust may be due mainly to the wear of automobile wheels [50]. Zn is solubilized primarily in the water used for cleaning the streets. Later, it can reach the rivers through the sewage system, increasing contamination by urban and road runoff [9].

**Table 2 ijerph-19-05263-t002:** A comparison of the heavy metal concentrations (mg kg^−1^) in street dust in Madrid and other selected cities from the literature review.

City	Populations	Pb	Zn	Cu	Ni	Cr	Cd	PL	Characteristic	Reference
Madrid	6,642,000	290	895	411	42	100	1.25	1889		Present paper
Murcia	453,238	137 229 163	673 618 693	157 234 271	35 76 31	80 169 93	-	1082 1326 1251	Low Medium High traffic	[47]
Madrid	3,000,000	1927	476	188	44	61	-	2696		[26]
Estambul	12,000,000	212	520	208	33	-	2.3	973 *		[25]
Islamabad	800,000	104	116	52	23	-	-	295 *		[24]
Dakha	140,921,000	74	154	46	26	-	-	300 *		[23]
Mexico City	20,000,000	116	447	93	45	93	-	794	Top urban soil	[22]
Oslo	1,762,000	180	412	123	41	-	1.4	756 *		[26]
París	2,137,000	1450	840	1075	25	-	-	3390 *	Highway	[49]
Monterrey	1,142,952	227	649	167	-	102	-	-		[48]
Teheran	12,183,391	56	46	74	-	-	-	-	Commercial use	[51]
background (asphalt)		22	61	14	11	13	-	121		Present paper

* Without considering the value of Cr which is not available.

In this study, the mean concentration of Pb was 290 mg kg^−1^ (Table 2), in contrast to the 1927 mg kg^−1^ reported by [26]. Gas, paint, and fine particle emissions from cars have historically been responsible for Pb pollution in cities, but since gasoline now has a low Pb content, pollution has decreased considerably (Table 2).

The average value of Cu in the dust from the street of Madrid (411 mg kg^−1^) can be considered high when compared with other cities, such as Monterrey, Mexico (167 mg kg^−1^) [48] or Tehran, Iran (74 mg kg^−1^) [49] (Table 2).

Based on the results obtained in this work, we can say that Cu represents a health risk, especially on busy streets. The stability of metal complexes tends to follow the following sequence: Cu > Fe > Mn = Co > Zn, so that in Cu, it can appear to form complexes, which can explain its persistence in the streets, unlike Zn and Pb.

The mean value of Ni in street dust was 42 mg kg^−1^. This element was reported in similar concentrations in other cities [2,4]; they are moderately anomalous values for natural soils in France [52] and close to the intervention level established by Adriano et al. [53].

The Cr values found in this work are higher than those obtained by [26] in Madrid 20 years ago and those found in the street dust of Murcia, another Spanish city. These Cr concentrations show that this element is increasing in the street dust of Madrid.

The toxicity of Cr depends not only on its concentration but also on its valence, so Cr^+3^ is slightly toxic compared with Cr^+6^. Given the oxidizing conditions for street dust, part of this heavy metal is in its maximum oxidation state. Regarding the concentration, Baize [52] indicates that natural soils in France with Cr levels between 150 and 534 mg kg^−1^ can be considered strongly anomalous. Similarly, in Madrid, the authorities should decide to mitigate contamination by this heavy metal, especially on the streets with the highest traffic density.

### 3.2. Environmental Pollution Index

The rates of environmental pollution due to urban dust in Madrid show, in general, that the levels of pollution are high or very high (Table 3). A higher CF of heavy metals means a higher risk for the local environment [54], where Pb was the element with a higher CF (89). The other heavy metals have very high values (>6), except for Ni, with moderate values. These levels are similar to Li et al. [55] for other cities.

Both the PLI and the RI are associated with the CF. The PLI provides an idea of the CF contribution of each metal to global pollution. In this study, the PLI reaches a value greater than 15; therefore, we deduced the maximum contamination level. The RI integrates the CF of each metal and the toxic-response factor for each metal, reaching values above 1200, which is considered a very high ecological risk.

The values of EF allow different levels of contamination to be determined, ranging from moderate enrichment in the case of Ni, substantial enrichment of Cr, very high enrichment of Cu, and extremely high enrichment of Cd, Pb, and Zn.

Several contamination levels were identified with the Igeo index, ranging from uncontaminated to moderately contaminated (Ni) to very highly contaminated (Pb), with the rest being moderate to highly contaminated.

Ni and Cr correlate at 0.907** (Cr = 1.57Ni + 33.979), which suggests that they come from the same source; this also happened in the city of Murcia [47], where the correlation was 0.991*.

The total pollutant load correlates with Pb (0.781**, PL = 1.147Pb + 1395.89), with Zn (0.747** PL = 1.61Zn + 463.423), and to a lesser extent with Cr and Cd (0.580** and 0.554**, respectively).

Cr and Pb could come from the lead chromate used in yellow paint, Zn is emitted by tires and additives to motor oil, and Cr can originate from exhaust emissions [56].

In short, the calculated indices show high or very high levels of anthropogenic contamination, especially by Cd, Pb, and Zn. Therefore, the authorities should identify the sources of these heavy metals to reduce the risk, as recommended by other authors [57].

### 3.3. Influence of Color and Density of Traffic

The dark color of coal and magnetite produced by fossil fuel burning is associated with heavy metals, mainly Pb and Mn, because they are used as ignition enhancers [58]. However, the lack of relationship between the color of the street dust samples and the Pb concentrations suggests that the primary sources are not cars. Taking actions that will substantially reduce the lead concentration in street dust will require more work to find the primary sources.

The density of traffic (DT) is the factor that most influences the concentration of heavy metals and the values of the environmental pollution indices (Figure 3). The concentration of Cd shows significant differences with the traffic density (*p* < 0.05), so that the lowest value (0.83 mg kg^−1^) is found in the intermediate traffic density, statistically significant for the high density of traffic (2.03 mg kg^−1^), but not significant for the lower one (1.37 mg kg^−1^).

Adriano et al. [53] defined a threshold concentration of Cd of 2 mg kg^−1^ in soils of parks and gardens as an intervention class; therefore, we recommend that the city authorities undertake corrective measures to reduce the concentration of Cd. Principally in those areas where traffic levels are equal to or greater than 40,000 cars per day, it is a common situation on the avenues of Madrid.

Cr and Ni present the highest concentrations on the busiest highways (between 40,000 and 80,000 vehicles per day) with a significance level of *p* < 0.01 and remain homogeneous in those with low and intermediate intensities (Figure 3A). For the rest of the heavy metals, traffic density is not a factor that allows differentiating the content of heavy metals in street dust (Figure 3B); therefore, they may come from other sources.

The mean value of Cr is 177 mg kg^−1^ in avenues with a traffic density of between 40,000 and 80,000 cars per day. This concentration of Cr is three times higher than that found by De Miguel et al. [26] in the same city, which indicates an apparent increase over these two decades and confirms the relationship between Cr and traffic levels, possibly due to the braking system. However, its origin could also be steel alloys, the plating industry, and other street furniture components.

Ni presented average concentrations of around 30 mg kg^−1^ for low and medium traffic density and 82 mg kg^−1^ at high ones, representing an almost double increase concerning the concentrations reported by De Miguel et al. [26] two decades ago. This increase in concentration, together with the dependence of this metal on traffic density, confirms its anthropogenic origin, where the most common sources are traffic, fertilizers, detergents, fuel burning, and the steel industry. In the presence of some organic complexing agents, Ni can form neutral or negatively charged soluble complexes, thus increasing its mobility compared with other metals and possibly contaminating aquatic ecosystems.

The CF, Igeo, and PLI pollution indices are sensitive to traffic density, whereas EF and RI do not show this behavior. In general, the maximum values have been observed in the highest traffic intensities. Thus, according to the Duncan test, CF has significantly different values in the case of Cd, Cr, and Ni (Figure 4), thus adopting a behavior similar to that shown by individual heavy metals. The highest values were obtained with Cd because this element is the most concentrated concerning its geochemical background, especially in the roads with the highest traffic density, where values of 38.4.

Contamination factors for Cr and Ni are slightly different from Cd, observing a direct relationship between traffic density and the index associated with these two elements (Figure 4). Both the Tukey test and the Duncan test establish two statistically different groups. On the one hand, the low and moderate traffic intensities have homogeneous and statistically lower CF values. On the other hand, the highest traffic density street shows the maximum values. The Igeo index behaves similarly to the pollution factor. Finally, PLI is statistically higher (*p* < 0.05) for the higher traffic intensities and remains homogeneous in the intermediate and low ones.

The PLI values in street dust calculated with CF of heavy metals contained in asphalt as a background value indicate that heavy metal contamination in urban dust is high throughout the study area; however, with the kriging indicator interpolation method, it was possible to spatially identify the most contaminated area (Figure 5).

The most polluted area largely coincides with the Central District, where points of interest such as Puerta del Sol and Plaza Mayor are located. As a tourist center of Madrid, it suffers from very interrupted traffic circulation of buses, taxis, motorcycles, and cars in a labyrinthine and hilly street layout. The least polluted area, located northwest of the study area, is in the Chamberí district, one of the wealthiest districts of the Spanish capital.

### 3.4. Human Health Index and Cancer Risk

For non-cancer effects, ingestion is the main route of exposure to dust particles and, therefore, the most significant health risk for children and adults for all metals. In second please is dermal contact, and inhalation was the route with the lowest values (Table 4). Similar results were found by Jadoon et al. [59] when studying the street dust of Kabul (Afghanistan). The HQ inh is between 10 and 100 times lower than the HQ ing and HQ derm, as in other previous works [41,43,53,55].

The HI values in urban dust in children varied from 0.016 (Cd) to 1.1 (Pb) and 0.0018 (Cd) to 0.11 (Pb) in adults. The HI values decreased in the following orders: Pb > Cr > Cu > Zn > Ni > Cd. The decreasing patterns of HIs for adults were similar; however, they were one order of magnitude lower.

Due to the HI values of heavy metals, there is no risk of developing adverse effects on human health (HI less than 1) for children and much less for adults (Table 4), who present even lower values, except in the case of Pb. A Pb value greater than 1 indicates a possibility of non-cancer effects occurring, with a probability that tends to increase as HI increases [38,42]; therefore, special attention should be paid to this heavy metal. Ingestion of Pb-contaminated dust is the main source of blood Pb in children. Children with a blood Pb level of ≥100 μgL^−1^ may manifest neurological and behavioral damage [59]. Moreover, metals may accumulate in fatty tissues of human bodies, presenting middle and long-term health risks, adversely affecting their physiological functions, disrupting the normal functioning of internal organs, or acting as cofactors in other diseases.

The HI values for Cr (4.3 × 10^−1^) and Cu (1.3 × 10^−1^), being higher than 1 × 10^−1^, are somewhat worrying since they can trigger many ailments [52].

For cancer risk, Cr, Cu, Ni, and Pb were evaluated by using the inhalation mode of exposure. CR in the street dust is summarized in Table 5. The overall cancer risks of metals decreased in the order Cr > Pb > Ni > Cd.

The authorities must make risk management decisions in those cases in which the cancer risk index is between 10^−6^ and 10^−4^. Our study’s values are below this range, so we do not consider risk [60,61]. However, our study had inherent uncertainties in estimating health risks from heavy metals exposure, such as actual duration of exposure and body weight. In addition, we did not consider other elements (Co, As, Al, Hg, Sn, Sr, etc.) nor different routes of exposure (indoor dust, contaminated soil, food, or water). Despite the uncertainties involved, the model proved to be a helpful tool for assessing human health risks and identifying the exposure pathways of most significant concern.

The wear of brakes, tires, car engines, and the wear of paint on building facades can be sources of Cr, Cu, Zn, and Pb, in addition to the gases and particles produced during combustion [56,59,62].

Due to the presence of heavy metals in street dust and the danger to health, we recommend preventing dust from entering houses (on shoes, through windows, etc.) and cleaning daily, mainly in homes with young children.

## 4. Conclusions

Zn is the primary heavy metal in street dust in Madrid, followed by Pb and Cu. Compared with values from two decades ago, Pb has dropped sharply, whereas Zn, Cd, and Cu have doubled, and Ni remains the same. We can say that the pollution levels are high or very high in Pb, Zn, and Cd regarding the environmental pollution indexes.

Traffic density is the factor that most influences the values of the environmental pollution indexes. The color did not work as an indicator of contamination.

The PLI distribution maps, with kriging interpolation, were able to spatially identify the most polluted area, which largely coincides with the Central District, to the southwest of the center of Madrid, with very interrupted traffic circulation.

In health risk assessment, ingestion is the main route of exposure to heavy metals in street dust for children and adults.

The health index value for children was 10 times higher than adults, confirming that heavy metals in street dust represent a serious potential risk to children’s health. Due to heavy metals, the carcinogenic risk for adults and children has values below 1 × 10^−6^, which means there is no risk for the population.

We consider that establishing the low emissions zone in Madrid has been a successful measure since the values of heavy metals in the street dust to which the citizens of this area were exposed were high. However, the next goal should be to identify the sources of Zn, Pb, and Cd to establish measures to reduce their concentrations in street dust.

## Figures and Tables

**Figure 1 ijerph-19-05263-f001:**
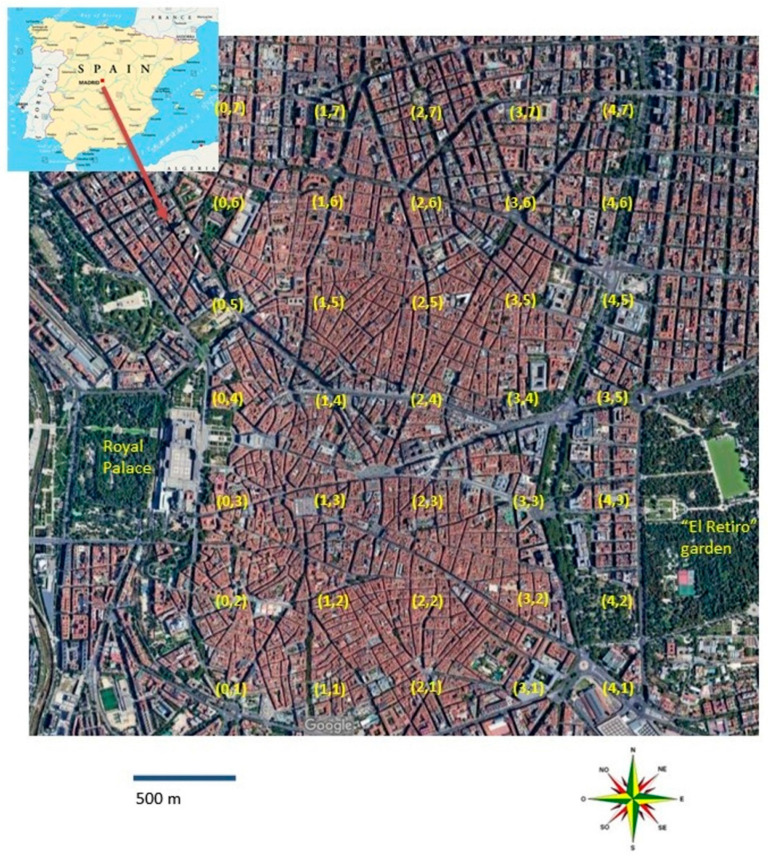
Map of Spain with the city of Madrid in the center of which it has been sampled Location of 35 street dust sampling sites in Madrid. Systematic sampling with 500 m between sites.

**Figure 2 ijerph-19-05263-f002:**
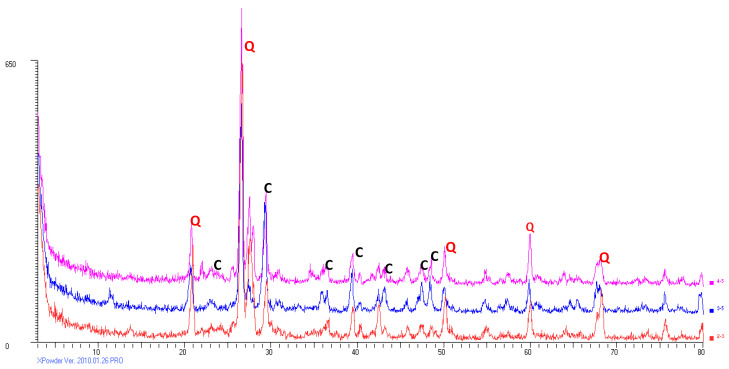
XR diffractogram of three samples of street dust. Q: quartz; C: calcite.

**Figure 3 ijerph-19-05263-f003:**
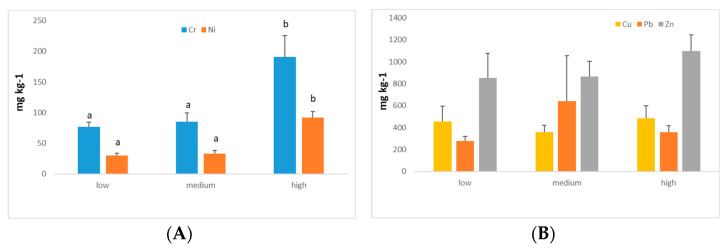
(**A**) Cr and Ni concentrations as a function of traffic density (low, medium, and high). Different lowercase letters indicate a significance difference among levels of traffic density for specific heavy metals (Tukey, *p* = 0.002 for Cr, and Tukey *p* = 0.03 for Ni). (**B**) Cu, Pb, and Zn concentrations as a function of traffic density (low, medium, and high). No statistical significance between different traffic densities for any of the heavy metals shown.

**Figure 4 ijerph-19-05263-f004:**
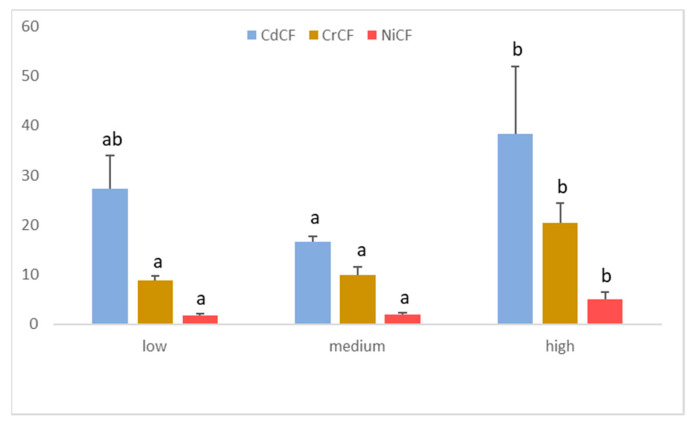
Variation of Contamination Factor (CF) as a function of traffic density (low, medium, and high) for Cd, Cr, and Ni. Different lower-case letters indicate a significance difference among levels of traffic density for specific heavy metals (CdCF; Duncan *p* = 0.044, CrCF; Tukey *p* = 0.002, and NiCF, Tukey *p* = 0.003).

**Figure 5 ijerph-19-05263-f005:**
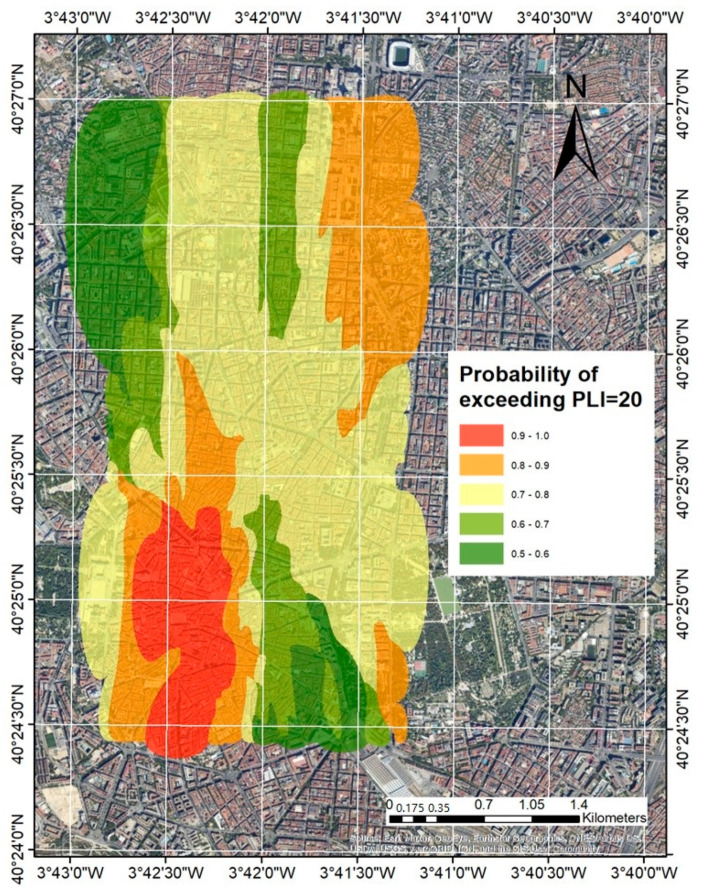
PLI spatial distribution map in the center of Madrid. The interpolation method was kriging indicator because it accepts data non-normality (Gaussian distribution) and converts the estimated values into indicator values that rank from 0 to 1. The output is the probability range of exceeding a cut-off value. We used a value of PLI = 20 as a cut-off value.

**Table 1 ijerph-19-05263-t001:** Reference doses (mg kg^−1^ day^−1^) (RfDing, RfDder, RfDinh) and cancer slope factor (mg kg^−1^ day^−1^) (SF) of heavy metals.

	Cd	Cr	Cu	Ni	Pb	Zn
RfDing	1.00E-03	3.00E-03	4.00E-0.2	2.00E-02	3.50E-03	3.00E-0.1
RfDder	1.00E-05	6.00E-05	1.20E-02	5.40E-03	5.25E-04	6.00E-02
RfDinh	1.00E-03	2.86E-05	4.02E-02	2.06E-02	3.52E-03	3.00E-01
SF	6.30E+00	4.20E+01		8.40E-01	4.20E-02	

**Table 3 ijerph-19-05263-t003:** Madrid environmental pollution indexes.

Index	Cd	Cr	Cu	Ni	Pb	Zn
CF	24.9	11.6	12.9	2.5	89.0	30.5
EF	43.5	19.1	21.9	4.2	157.2	53.0
Igeo	3.8	2.7	2.7	0.4	5.6	4.0

**Table 4 ijerph-19-05263-t004:** Mean non-cancer index values for children and adults: Ding, Dinh, and Dderm (mg kg^−1^ day^−1^) and Hazard Quotient for each element and exposure route.

Children							
Element	Ding	Dinh	Dderm	HQing	HQinh	HQderm	HI
Cd	1.6E-05	4.5E-10	2.5E-08	1.6E-02	4.5E-07	4.5E-05	1.6E-02
Cr	1.3E-03	3.6E-08	2.1E-06	4.3E-01	1.3E-03	6.0E-04	4.3E-01
Cu	5.3E-03	1.5E-07	8.4E-06	1.3E-01	3.7E-06	1.2E-05	1.3E-01
Ni	5.3E-04	1.5E-08	8.6E-07	2.7E-02	7.3E-07	2.8E-06	2.7E-02
Pb	5.6E-03	1.6E-07	9.0E-06	1.6E+00	4.5E-05	3.0E-04	1.1E+00
Zn	1.1E-02	3.2E-07	1.8E-05	3.8E-02	1.1E-06	5.4E-06	3.8E-02
**Adults**							
**Element**	**Ding**	**Dinh**	**Dderm**	**HQing**	**HQinh**	**HQderm**	**HI**
Cd	1.7E-06	1.6E-10	5.2E-09	1.7E-03	4.5E-07	4.5E-05	1.8E-03
Cr	1.4E-04	1.3E-08	4.2E-07	4.6E-02	1.3E-03	6.0E-04	4.8E-02
Cu	5.6E-04	5.3E-08	1.7E-06	1.4E-02	3.7E-06	1.2E-05	1.4E-02
Ni	5.7E-05	5.4E-09	1.7E-07	2.9E-03	7.3E-07	2.8E-06	2.9E-03
Pb	6.0E-04	5.7E-08	1.8E-06	1.7E-01	4.5E-05	3.0E-04	1.1E-01
Zn	1.2E-03	1.2E-07	3.7E-06	4.1E-03	1.1E-06	5.4E-06	4.1E-03

**Table 5 ijerph-19-05263-t005:** Mean values for Cancer Risk for each element.

CdLADD	CrLADD	CuLADD	NiLADD	PbLADD	CdCR	CrCR	NiCR	PbCR
9.3E-11	7.5E-09	3.1E-08	3.1E-09	3.3E-08	5.9E-10	3.2E-07	2.6E-09	1.4E-09

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
