# Peer review of "Estimation of Ecological and Human Health Risks Posed by Heavy Metals in Street Dust of Madrid City (Spain)"

_ijerph, 2022, doi:10.3390/ijerph19095263_

Round 1

Reviewer 1 Report

Comments to the Editor

Title: “Estimation of ecological and human health risk by heavy met- 2 als of street dust of Madrid city (Spain)

A lot of mistakes need to improve under the Major Correction

Positive aspects of the manuscript…

Abstract: metals full forms missing.

  1. Carcinogenic risk (CR) details missing in the method part.
  2. Rapids analysis or seasonal variation observed? The author failed to give,
  3. Spatial distribution patterns of heavy metals need to be added more techniques.
  4. Study areas need to be revised without errors.
  5. Are sample collection details missing- like a core sample?
  6. 2.3. Environmental Pollution Index and the Potential Risk treet dust analysis study Area and 137 Sampling – not clear- need to be revised the sub title.
  7. 2.3.2. Pollution Load Index (PLI)
  8. 2.3.3. Pollution Load Index (PLI) need to be rechecked the title and statement – missing leading the information – more confusion. Need to be clarified.
  9. What are the metal analysed in the XRD? And XRD makes and model details missing.
  10. ICP-MS standards details missing.
  11. ICP-MS make and model details missing.
  12. Figure legends need to be revised (Fig 1) with satellite images in Spain representation.
  13. Few more analyses like PCA may be added in the revised version for source of heavy metal in the study site.
  14. Source of pollution point details missing.
  15. Study site sampling point Lat and long may give in table format in supplementary.
  16. Method part:
  17. Cancer Risk details need to be added more still lacking – briefly.
  18. Why author taken only three samples in XRD analysis for any specific reason?\
  19. Table 2 need to be clarified- on top of the column stating that present study, similarly way presented in the bottom also which one? Need to correct it.
  20. Author may follow the spatial distribution of metals as published by DOI: 10.1016/j.rsma.2020.101242. May give a clear picture with other techniques  (Kriging interpolation technique use for Mapping (Sachithanandam et al., 2020).
  21. Table 4 data need to be improved result and discussion part – briefly.
  22. However, a few corrections on the statistical analysis part are still required and recent references should be added in the revised version.
  23. I strongly recommend this manuscript be accepted for publication in Sustainability with Major corrections mentioned below:
  24. In addition to that, to understand the relationship between these parameters, it is suggested that authors may consider identifying regression equation.
  25. Conclusions part - need to be revised in briefly.

General remarks on the manuscript:

  1. The typographical errors should rectify throughout the manuscript.
  2. The study area needs to be revised with good quality satellite data.
  3. Syntax error should rectify it?

Author Response

The authors are grateful for all your recommendations which have improved the quality of this paper. We have tried to follow all your suggestions.

Please find attached the file with the answers to each of your suggestions.

Best regards

Reviewer 2 Report

The goal and results of this study seems to be interesting and good. However, the followings should be revised, I think. 

  1. I suggest presenting results and considerations separately.
  2. Equation 8 should be corrected.
  3. It would be better provide the correlation among Contamination Factor (CF), Enrichment Factor (EF), Geo-accumulation Index (Igeo), Potential Ecological Risk Index (RI), Pollution Load Index (PLI), and the Human 
    Health Index.
  4. Above all, it is necessary to compare with previously published similar studies. For example, this research should present the advanced points and limitations compared to previous studies.
  5. Lastly, there is an awkwardness of the phrase, so please get an English correction by native speaker. 

Author Response

(The authors gave the same response as above.)

Round 2

Reviewer 1 Report

Few Typogrphic error has to rectify before going to publication.